palaeontology, taxonomy and systematics

morphology, palaeozoic, organic-walled microfossils, chitinozoans, biologic affinity

**Authors for correspondence:**
Yan Liang
email: liangyan@nipas.ac.cn
Olle Hints
email: olle.hints@taltech.ee

# Morphological variation suggests that chitinozoans may be fossils of individual microorganisms rather than metazoan eggs

Yan Liang[1,3,4], Joseph Bernardo[5,6,7,8], Daniel Goldman[9], Jaak Nõlvak[3], Peng Tang[2,4], Wenhui Wang[10] and Olle Hints[3]

[1]CAS Key Laboratory of Economic Stratigraphy and Palaeogeography, and [2]State Key Laboratory of Palaeobiology and Stratigraphy, Nanjing Institute of Geology and Palaeontology, Chinese Academy of Sciences, Nanjing 210008, People's Republic of China
[3]Department of Geology, Tallinn University of Technology, Tallinn 19086, Estonia
[4]Center for Excellence in Life and Paleoenvironment, Chinese Academy of Sciences, Nanjing 210008, People's Republic of China
[5]Marine Biology Interdisciplinary Program, [6]Program in Ecology and Evolution, [7]Laboratory of Evolutionary Ecology and Climate Change and [8]Department of Biology, Texas A&M University, TX 77843, USA
[9]Department of Geology, University of Dayton, Dayton, OH 45469, USA
[10]Key Laboratory of Metallogenic Prediction of Nonferrous Metals and Geological Environment Monitoring, Ministry of Education, School of Geosciences and Info-Physics, Central South University, 410083 Changsha, People's Republic of China

(iD) YL, 0000-0002-2376-8202; JB, 0000-0002-5516-4710; JN, 0000-0002-6678-9919; WW, 0000-0002-4251-0653; OH, 0000-0003-4670-4452

Chitinozoans are organic-walled microfossils widely recorded in Ordovician to Devonian (*ca* 485–359 Mya) marine sediments and extensively used in high-resolution biostratigraphy. Their biological affinity remains unknown, but most commonly, they are interpreted as eggs of marine metazoans. Here, we provide new insights into their palaeobiology from three lines of inquiry. We examine morphological variation of a new, well-preserved Late Ordovician species, *Hercochitina violana*; analyse a compiled dataset of measurements on 378 species representing all known chitinozoan genera; and compare these data with the size variation of eggs of both extinct and extant aquatic invertebrates. The results indicate that the magnitude of size variation within chitinozoan species is larger than observed in fossil and modern eggs. Additionally, delicate morphological structures of chitinozoans, such as prosome and complex ornamentation, are inconsistent with the egg hypothesis. Distinct and continuous morphological variation in *H. violana* is more plausibly interpreted as an ontogenetic series of individual microorganisms, rather than as eggs.

## 1. Introduction

Chitinozoans are an extinct group of cryptic organic-walled microfossils widely distributed in Ordovician to Devonian marine sedimentary rocks (*ca* 485–359 Myr old). They are characterized by a radially symmetrical shell, known as vesicle, which is commonly jar-, vase- or tube-shaped and has an opening in one end sealed with a plug (prosome) or lid (operculum). The vesicle length varies from *ca* 50 to 2000 µm, with a newly reported largest one of 2700 µm [1], mostly common between 100 and 500 µm.

Chitinozoans originated (leaving out the problematic record in Cambrian [2]) and diversified in the Great Ordovician Biodiversification Event [3,4], survived the end Ordovician extinction [5] and disappeared during the Late Devonian extinction [6]. Except for their enigmatic life history, they have attracted wide attention

as a tool in high-resolution biostratigraphy [7–9], providing insights for several key events [10–12]. Yet the biological affinity of chitinozoans remains unknown. Since their first description in the 1930s [13], multiple hypotheses [14] have been put forward trying to explain their biological origin and ecology, including classification as protists, metazoans or egg capsules of metazoans. Later, a fungi hypothesis [15], possible polyphyletic origin [16], vegetative reproduction [17] and ontogenetic development [18] have been hypothesized, however, with less general acceptance. Over the last three decades, chitinozoans have almost exclusively been interpreted as eggs of unknown marine metazoans [12,19,20], possibly some worm-like animals [21]. No further discussions related to their affinity have been advanced since then, even though the egg hypothesis makes no allowance for, nor does it explain, some of the morphological structures found in Chitinozoa.

Here, we re-evaluate the egg hypothesis from the perspective of morphological variation of chitinozoans. In order to allow for a more quantitative analysis, a detailed case study focusing on single highly variable species *Hercochitina violana* sp. nov. is conducted. Additionally, we compiled a dataset of measurements of 378 randomly selected species representing all 57 chitinozoan genera currently known, as well as a dataset of coefficients of variation (CV) in eggs from 45 extant aquatic metazoan species. For the first time, this allowed a reliable comparison of the magnitude of size variation between chitinozoans and undisputed eggs to further assess their possible analogy.

## 2. Material and methods

Well-preserved specimens of *H. violana* sp. nov. were collected from six limestone samples in the upper part of the Viola Springs Formation at the auxiliary Global Stratotype Section and Point section for the base of the Katian Stage (Upper Ordovician) in Oklahoma, USA [22]. Rock samples were processed following the standard palynological techniques [23] or digested in diluted acetic acid, without any further process for oxidation. All specimens were studied and photographed using optical microscopy. Selected specimens were observed and photographed using scanning electron microscopy (SEM) for a further detailed examination of morphology. A total of 331 three-dimensionally preserved specimens of *H. violana* sp. nov. obtained from two of the six samples were measured for quantitative analysis (electronic supplementary material, S1). All specimens are stored at the Nanjing Institute of Geology and Palaeontology, Chinese Academy of Sciences.

A compiled chitinozoan dataset includes 593 records of all 57 type species and additional 321 species randomly selected, without previous sorting or emendation, from the literature, has been established and presented in the electronic supplementary material, S2. The CV of egg sizes representing six phyla and 45 species of extant aquatic (mostly marine) metazoan species was assembled and the criteria for selecting the CV data are detailed illustrated in the electronic supplementary material, S3. Multivariate statistical analysis (principal component analysis (PCA)) and locally weighted scatterplot smoothing (LOESS) regression were carried out using R [24].

## 3. Results

### (a) Systematic palaeontology

Incertae sedis group CHITINOZOA Eisenack, 1931
 Order PROSOMATIFERA Eisenack, 1972

Family CONOCHITINIDAE Eisenack, 1931, emend. Paris, 1981
 Subfamily BELONECHITININAE Paris, 1981
 Genus *Hercochitina* Jansonius, 1964
 *Hercochitina violana* sp. nov. Nõlvak and Liang
 (figure 1a–k)
 ?1985 *Hercochitina repsinata* (Schallreuter) n. comb; Melchin & Legault, p. 208, pl. 1, fig. 1.
 2007 *Clathrochitina*? sp. n. 1; Goldman *et al.*, p. 267, fig. 13: 1, 4.

**Etymology**. In reference to the type stratum, the Viola Springs Formation of Oklahoma.

**Diagnosis**. This species is characterized by its distinct ornamentation, low and multi-rooted longitudinal spines developed in the lower part of the chamber, and jungle-like complex ones on the basal margin. A constriction near the base and a mucron on the base are also evident.

**Types**. Holotype, sample HWY99-43 (43 m above the base of the formation), figure 1i (repository number NIGPAS19-28); Paratype, sample HWY99-37.8, figure 1a (repository number NIGPAS19-20).

**Type locality**. Section D in the west side of State Highway 99, 3.3 miles south of Fittstown, Arbuckle Mountains, south-central Oklahoma, USA [22].

**Type stratum**. The upper part of the Viola Springs Formation, ranging from sample HWY99-37.8 to HWY99-51, Katian, Late Ordovician.

**Material**. Seventy-three specimens from 50 g of sample HWY99-37.8, 258 specimens from *ca* 300 g of sample HWY99-43 were measured.

**Dimensions**. See figure 2 and the electronic supplementary material, S1.

**Description**. The vesicle size ranges from small to medium and the outline from stout to slender. Chamber sub-cylindrical to subconical with a short cylindrical neck developing a short flaring collar. The flexure is so broad that it is difficult to identify the boundary between the neck and the chamber. Flanks are straight to slightly convex. Basal margin sharp to blunt. Base is flat with a mucron. Constrictions are inconspicuous or weak in smaller specimens and become stronger and more distinct in larger, always slender, specimens. The constriction diameter is about three-fifths to nine-tenths of the maximum diameter at the base of the vesicle. The vesicle wall is covered by multiple kinds of spines: simple or multi-rooted spines distributed near the collar, low and multi-rooted longitudinal spines developed in the lower part of the chamber around the constriction, much stronger and more complex ones on the basal margin, and simple and multi-rooted spines developed concentrically in the base around the mucron.

**Remarks**. The new species, *H. violana* sp. nov., is easily distinguished from other *Hercochitina* species by its less-developed longitudinally distributed spines in the upper and middle parts of the chamber. The distinct ornamentations more likely represent a transitional form between the typical *Belonechitina* and *Hercochitina*, but more closely resemble the ornamentation of *Hercochitina* by occupying well-developed vertical rows of spines or distinct crests, even if these distinguished features are confined within the lower chamber of the vesicle, which is the main feature of typical *Hercochitina*.

The species resembles *Hercochitina repsinata* (Schallreuter) Melchin & Legault [25] both in vesicle size, outline and

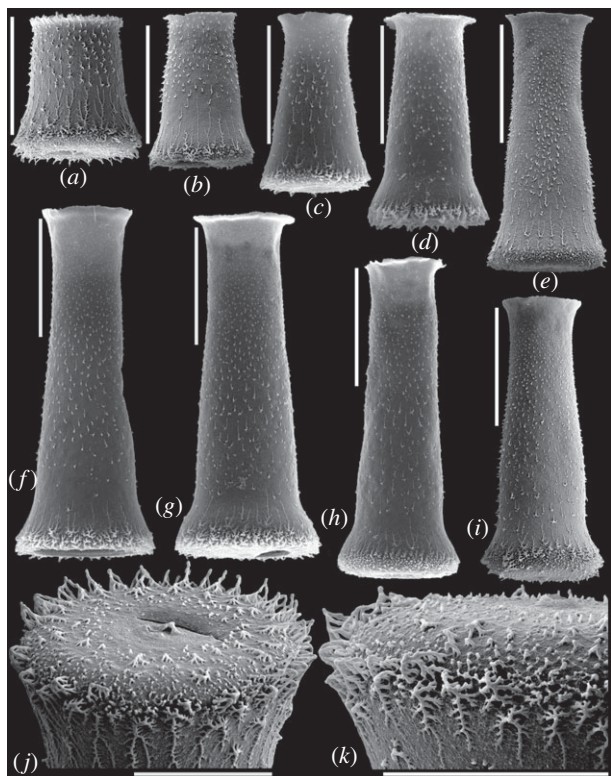

**Figure 1.** SEM images of *H. violana* sp. nov. showing the morphologic variation. (*a,b,e,i*) Specimens from sample HWY99-37.8: (*a*) NIGPAS19-20, (*b*) NIGPAS19-21, (*e*) NIGPAS19-24, (*i*) NIGPAS19-28. (*c,d,f–h*) Specimens from sample HWY99-43: (*c*) NIGPAS19-22, (*d*) NIGPAS19-23, (*f*) NIGPAS19-25, (*g*) NIGPAS19-26, (*h*) NIGPAS19-27. (*j,k*) Enlarged images of (*a*) and (*i*) showing the distinct spiny ornamentation on the vesicle margin and the mucron at the middle of the base. Scale bars of (*j*) and (*k*) represent 50 µm, others represent 100 µm.

similar ornamentation. It differs from *Belonechitina repsinata* Schallreuter, 1981 [26] in developing a remarkable constriction near the base. The holotype of *B. repsinata* is considered to be a badly preserved *Belonechitina robusta* (Eisenack, 1959) [27]. Thus, *B. repsinata* might be an invalid name and the specimens reported by Melchin & Legault [25] could potentially belong to *H. violana* sp. nov. Additionally, the specimens previously identified as Gen. et sp. n. 1 (*Clathrochitina*?) [22] are from the same level of the same section as the materials recovered in the present study. They are reassigned to *Hercochitina* because the spines on the margin are multi-rooted rather than processes and the flexures are inconspicuous.

## (b) Morphological variation of *Hercochitina violana* sp. nov

The morphology of the new species shows variation both in vesicle size and outline (figures 1 and 2). The vesicle length (L in abbr.) varies more than a threefold from 94 to 317 µm. The maximum diameter ($D_P$) and the diameter of the constriction ($D_{cons}$) vary from 54 to 125 µm and 40 to 94 µm, respectively. All the measured parameters are normally distributed and the specimens on the PCA are well grouped, which support the idea that all the specimens belong to a single population.

Overall, the length variation is much greater than the diameter variation (figure 2*g*). The vesicle outline changes from a stout shape to a slender form with an indistinct but

longer neck. The constriction also becomes more distinct as the specimens become larger and slenderer. Taking into consideration the normally distributed parameters, the substantial but continuous variation suggests a longitudinal growth trend rather than intraspecific variation. If the specimens are exhibiting population variation, it is plausible to argue that the predominant length, from *ca* 150 to 210 µm, potentially represent the mature stage of the species. Individuals longer than 210 µm and shorter than 130 µm show significantly decreased proportionality and may represent the largest grown and juvenile specimens, respectively. Notably, except for some small-chambered taxa of the family Desmochitinidae [22], such as *Desmochitina*, *Calpichitina* and *Eisenackitina*, no specimens smaller than the smallest *H. violana* sp. nov have been reported from the same sample.

## 4. Discussion

### (a) Variation of other chitinozoans

According to the compiled dataset, including 574 valid records (measurement data of the other 19 cases are incomplete; in the electronic supplementary material, S2), the ratio between the longest and shortest specimens within the measured assemblage ranges from 1.03 to 4.43, three-quarters of which range from 1.0 to 2.0 (figure 3*a*); however, in most species, only a relatively small number of specimens have been measured. When using data with more than 40 specimens measured, the ratio increases: 74% of the 116 valid data range from 1.5 to 2.5 and 19% with an $L_{max}/L_{min}$ ratio more than 2.5 (figure 3*b*). Furthermore, 60% of the data share a ratio around 2.5 and only 15% are less than 2 but still larger than 1.5 when the ratios are based on more than 100 specimen measurements (figure 3*c* with 20 valid data).

The LOESS regression of 498 valid data (complete records with measured specimen numbers) indicates that variation estimates increase when at least 40 specimens are measured and an expected ratio around 2.7 is estimated (figure 3*d*). The variability range in the family Desmochitinidae (ratio of *ca* 2.7) has no evident difference from the Conochitinidae (*ca* 2.8) and Lagenochitinidae (*ca* 2.5) in the expected data by LOESS regression (figure 3*e–g*), even though the size of the dataset within each family varies significantly. These results indicate that Desmochitinidae, characterized by the key structure of operculum and with the only record of cocoon-like preservation [19,28]—several tens of specimens clustered together and covered by an organic-walled sheath—shares a similar morphological variation with other chitinozoans which possess a prosome instead of an operculum.

However, an analysis focusing on the data gathered on *Desmochitina* specimens, which are characterized by an ovoid vesicle and absence of neck, and sealed with the most typical operculum, indicates a smaller morphological variation. Except for the record of *Desmochitina ornensis*, some specimens of which occupying a short but distinct neck as shown in the type locality [23], all the other 29 datasets share an $L_{max}/L_{min}$ ratio limited to 2 (figure 4*a*) and two-thirds range from 1.2 to 1.6. The LOESS regression, which might be less reliable owing to the limited data, shows an estimated value around 1.9 and 1.5, respectively, with or without *D. ornensis* (figure 4*b*). The variation of *Desmochitina minor*, within which recorded the only cocoon-like

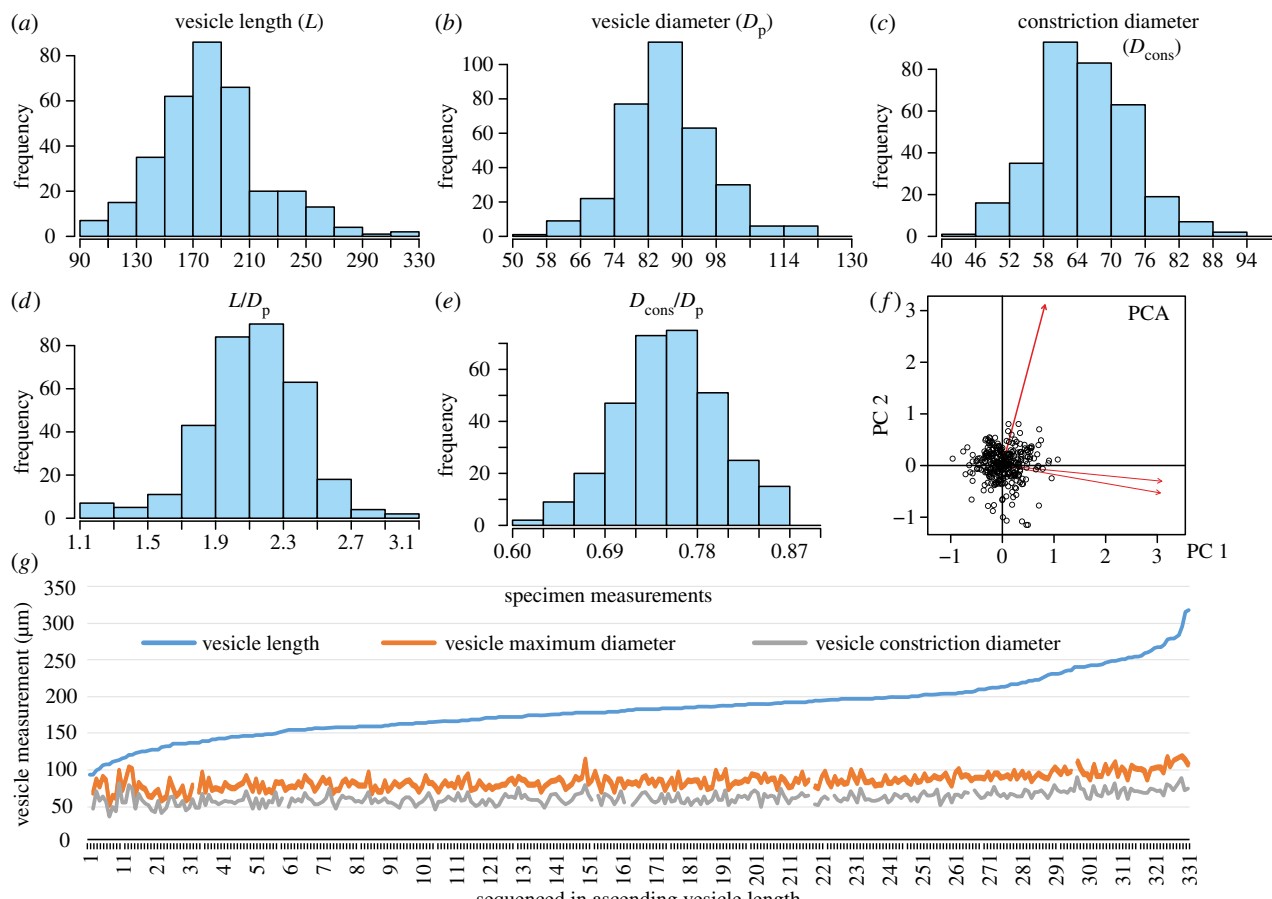

**Figure 2.** Histograms of measured parameters. (*a*)–(*e*) Distributions of different parameters of *H. violana* sp. nov. (*f*) PCA result based on data of vesicle length, maximum and constriction diameter. (*g*) A general view of the variation of vesicle length, maximum and constriction diameter which indicates a longitudinal growth pattern. (Online version in colour.)

preservation, ranges from 1.1 to 1.9 with an average value of 1.47 based on 11 datasets in which the number of measured specimens varied from 2 to 50. Although the data of *Desmochitina* are limited for statistical analysis, a plausible trend is that taxa within *Desmochitina*, especially the *D. minor*, sharing an $L_{max}/L_{min}$ ratio of less than 2 can be estimated, which is smaller than the variation in other taxa occupying a neck.

Except for the group of *Desmochitina*, on average, the intraspecific chitinozoan vesicle length variation is around 2.7 and can be more than 3 when a sufficient number of specimens are measured. This new analysis of variation is important because the size limit is among the main diagnostic characters used in species definitions and identifications. However, the biological variation in fossil populations has always been difficult to document because of preservation, time-averaging and inability to assess geographical variation. In addition, chitinozoan specimens are commonly compressed and deformed, the diagnostic ornamentations are easily removed or destroyed and have a relatively simple anatomical outline. All of this has contributed to the fact that the range of variability in chitinozoans has often been underestimated, and promoted a general hesitancy for workers to consider a group of specimens with great size variation to belong to a single species. In other words, size variation is potentially even larger within most groups in chitinozoans.

## (b) Variation in fossil and extant invertebrate eggs

Fossils of aquatic invertebrate eggs are generally rare [29–35] and little is known about their size variation owing to limited

numbers of specimens and often poor preservation. Almost all reported fossil eggs of aquatic invertebrates have a simple rounded or ovoid shape without any plug or lid-covered opening. They are different from chitinozoans both in outline and main features. The reported ratio of $L_{max}/L_{min}$ is generally below 2 [34,36,37], which is smaller than the estimated size variation of chitinozoans when enough specimens were measured, with the exception of *Desmochitina*. Overall, the fossil egg-size variation is poorly known and has thus little significance in the analogous study owing to the limited number of case studies as well as limited number of eggs in each individual case.

However, the size variation of eggs is well studied among a wide phylogenetic range of extant animals [38–42]. Therefore, we compiled a dataset of 50 CV estimates, representing six phyla, 10 classes, 20 orders and 45 species of extant aquatic (mostly marine) metazoan species (electronic supplementary material, S3.2), to make a comprehensive view of egg-size variation. The distribution of these values is normal (Shapiro–Wilk $W$-test; $W = 0.916863$, $p < 0.0018$; figure 5) with a mean of 6.59, a standard error of 0.590, a median of 5.35, and a range of 0.7–16.67. The upper and lower 99% confidence intervals are 8.23 and 5.00, respectively.

However, the CV of size variation on *H. violana* is 20.68, with an $L_{max}/L_{min}$ ratio of 3.4. A one-tailed, one-sample $t$-test [43] indicated that the CV of *H. violana* is significantly different from the above distribution ($t = 3.344$, $p < 0.001$). This allows us to conclude that extant aquatic invertebrates in our broad phylogenetic survey exhibit smaller variation

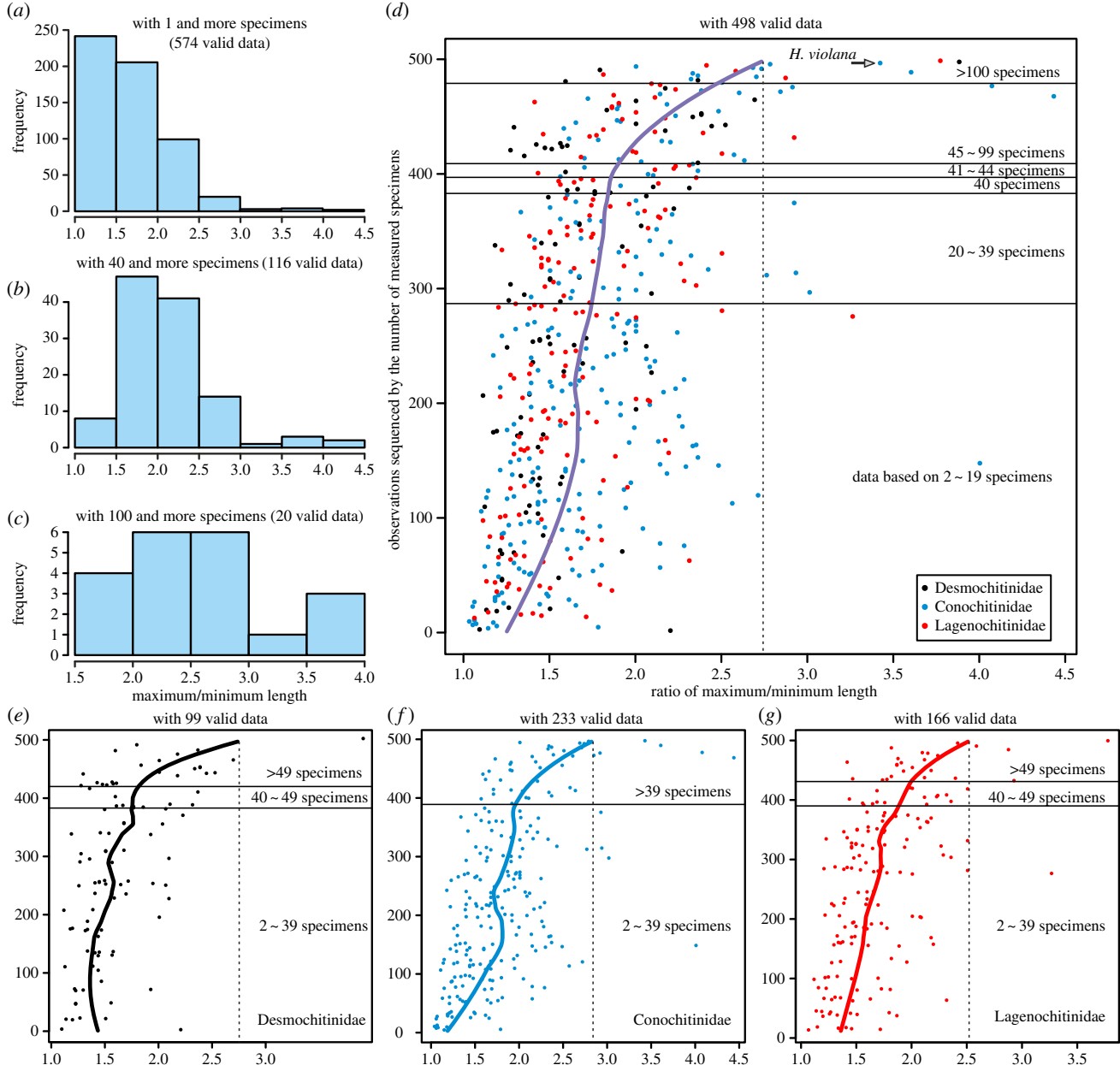

**Figure 3.** Size variation of chitinozoans. (*a–c*) Histograms of the maximum to minimum length ratio based on published data showing that the ratio increases as sample size (number of individuals measured) increases. (*d*) LOESS regression showing an increase in ratio as more specimens were measured. (*e–g*) LOESS regression of the three families with an increased ratio marked by the transversal lines. (Online version in colour.)

in egg size than exists in chitinozoans. The *t*-test rejected the hypothesis that the CV of *H. violana* reported here is part of the same distribution with a high level of probability. In other words, data on the egg-size variation of both fossils and extant invertebrates are incompatible with the hypothesis that chitinozoans are eggs.

## (c) Morphological implications on the biological affinity

Apart from the distinct size variation of chitinozoans, the predominant egg hypothesis does not explain well the complex structure of prosome, which is the primary character for subdividing chitinozoans at the order level. For example, systematic studies of prosome morphologies have documented diverse forms and structures among chitinozoan species [44,45]. Another detailed SEM image of a prosome [14] shows a delicate and complex structure consisting of several transverse septa connected by a central vertical structure. It is remarkable to see that there is a thin membrane-like structure

around the septa and a wide membrane covering the lowermost thick septum, the rica. This structure is much more complex than is known in any eggs. It is more likely a key functional structure for the chitinozoan individual, even though the specific function remains hitherto unknown. Also, the strong processes and complicated spines developed in some chitinozoans, together with the rapid evolutionary trend, especially in their ornamentation [19,25,46], are atypical of metazoan eggs.

In the history of exploring the biological affinity of chitinozoans, three main stages can be summarized: (i) relationship with various groups of Protozoa mainly based on the vesicle outline from the 1930s to 1980s; (ii) eggs of unknown metazoans, ignited by the discovery of the cocoon-like clusters [28] during the 1960s to 1980s; and (iii) widely adopted as eggs or egg capsules of an unknown group of soft-bodied metazoans since the 1990s. Detailed information on different hypotheses can be found in the literature [14,19,23,47]. The success of the egg

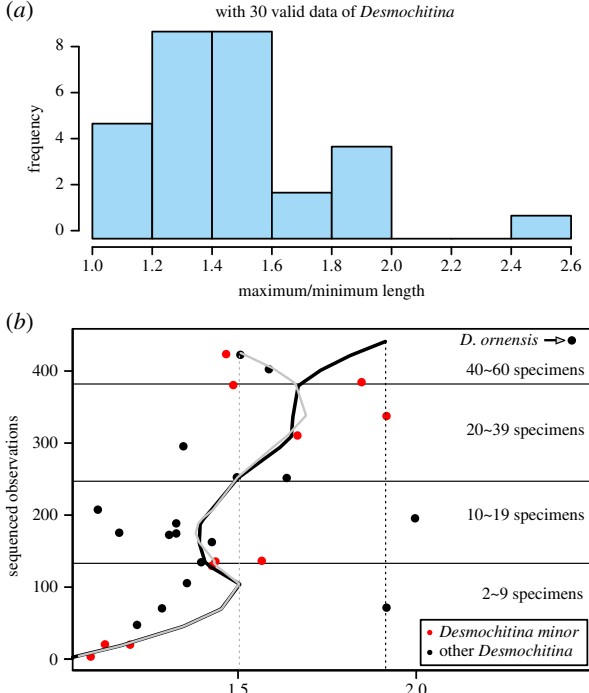

**Figure 4.** Size variation of *Desmochitina*. (*a*) Histogram distribution of the maximum to minimum length ratio based on 30 published data of *Desmochitina*. (*b*) LOESS regression based on the data of *Desmochitina*. The black and grey curves are based on the data with or without the datum of *D. ornensis*, respectively. (Online version in colour.)

hypothesis is mainly owing to the hermetically sealed vesicle which restricts communication between the inner chamber and the outer environment, and the cocoon-like clustered specimens. However, among the 57 recognized chitinozoan genera [48,49], the cocoon-like clusters have only been observed in a single genus—*Desmochitina* [19,28]. Furthermore, the cocoon-like preservation could also make sense to be interpreted as a reproduction stage, for instance, a multiple fission stage of an individual organism.

In addition to the cocoon-like clusters in *Desmochitina*, there are linear or coiled catenary forms, which are common in *Desmochitina*, *Bursachitina*, *Armoricochitina* and *Linochitina* [17,50]. Those forms have also been compared to the eggs of polychaetes and molluscs [16,23,51]. However, the difference is that the eggs in similar catenaries lack a covered opening sealed by a prosome or operculum, which is the diagnostic character of chitinozoans. Besides, the various types of attachments seem to be not suitable to be evaluated for the biological affinity significance because many invertebrates display a similar mode of egg laying [21].

## 5. Conclusion

The results show that the range of variation in most chitinozoans is much larger than is typical of eggs of extant marine invertebrates or the hitherto reported undisputed fossil eggs. The range and type of size variation shown in *H. violana* sp. nov. potentially indicate an ontogenetic series rather than an intraspecific variation. Additionally, there is no analogue in extant metazoan eggs which have such complicated and delicate structures as developed in chitinozoans, not to mention the rapid evolutionary trend of ornamentations, which is also inconsistent with the prevailing egg hypothesis. It is more plausible to argue that most chitinozoans, with the exception

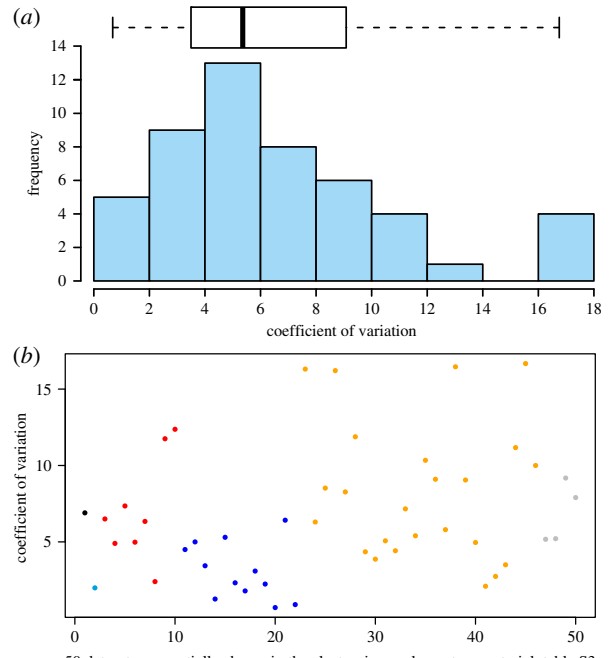

**Figure 5.** The CV of extant aquatic egg-size distributions. (*a*) Histograms and box-and-whisker plot of the CV values. (*b*) A plot of the CV of all the 50 datasets. Different colours indicate different phyla as follows: black, Bryozoa; cyan, Annelida; red, Mollusca; blue, Arthropoda; orange, Echinodermata; grey, Chordata. (Online version in colour.)

of the unique *Desmochitina*, represent independent organic-walled microorganisms rather than eggs of metazoans.

Finally, we note that because chitinozoans are an extinct group, there remains the unlikely possibility that they represent the eggs of an extinct group of marine organisms that displayed a uniquely large size variation in addition to exhibiting very unusual structures for eggs. We think that this is an extremely unlikely hypothesis and is contrary to the basic concepts of uniformitarianism. Unless demonstrated to the contrary, it is reasonable to assume that the range of variation exhibited in the eggs of modern and available fossil invertebrate metazoans is also representative of the rest of Earth's past biota.

Furthermore, some of the previously reported abnormal specimens, such as the *Conochitina* ex aff. *campanulaeformis* [52], *Conochitina* sp. [53] and *Ancyrochitina* cf. *brevis* [20], may provide some new insights to question the egg hypothesis. All of those abnormal specimens differ from the real malformed specimens [12,20] in occupying regularity and repeatability, i.e. all the 'malformation' appeared at the base of a 'normal' specimen and highly resemble the 'normal' individual. The 'malformed' specimen seemed to have the potential to develop into a complete specimen, which point to a reproduction stage as previously suggested by Cramer & Díez [17]. However, owing to the limited specimens, the reproduction hypothesis requires further proof and has the potential to further decode the nature of these enigmatic microorganisms.

Data accessibility. This article has no additional data.
Authors' contributions. Y.L. designed the study, carried out the microfossil laboratory work, participated in systematic identification, data analysis and drafted the manuscript; J.B. conducted the egg size literature review, participated in the statistical analyses and drafted the manuscript; D.G. collected field data and revised the manuscript; J.N. carried out the microfossil laboratory work and participated in systematic identification; P.T. and W.W. critically revised the manuscript; O.H. coordinated the study and critically revised the

manuscript. All authors gave final approval for publication and agree to be held accountable for the work performed therein.

Competing interests. We declare we have no competing interests.

Funding. This work was supported by the National Natural Science Foundation of China (grant nos. 41602010, 41772001); the Estonian Research Council (grant nos. MOBJD62, PUT611); and the Strategic Priority Research Program of Chinese Academy of Sciences (grant no. XDB26000000).

Acknowledgements. We thank Renbin Zhan and Chenyang Cai for providing constructive suggestions, and Florentin Paris for sharing the chitinozoan database. This paper is a contribution to the IGCP Project 653 'The Onset of the Great Ordovician Biodiversification Event'.

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
