## [Reviewer comments · Proceedings of the Royal Society B: Biological Sciences]

Review History

RSPB-2019-1270.R0 (Original submission)

Review form: Reviewer 1

Recommendation

Accept as is

Scientific importance: Is the manuscript an original and important contribution to its field?
Excellent

General interest: Is the paper of sufficient general interest?
Excellent

Quality of the paper: Is the overall quality of the paper suitable?
Excellent

Is the length of the paper justified?
Yes

Should the paper be seen by a specialist statistical reviewer?

No

Do you have any concerns about statistical analyses in this paper? If so, please specify them explicitly in your report.

No

It is a condition of publication that authors make their supporting data, code and materials available - either as supplementary material or hosted in an external repository. Please rate, if applicable, the supporting data on the following criteria.

Is it accessible?

N/A

Is it clear?

N/A

Is it adequate?

N/A

Do you have any ethical concerns with this paper?

No

Comments to the Author

I revised this paper for another journal, where it was (apparently) rejected.

It is not surprising that some of the chitinozoan workers want to stick to their "egg hypothesis" and do not see this paper in a good light.

However, the paper presents a very well argued study, and with clear evidence they propose another scenario.

My congratulations for a fantastic paper!

I strongly recommend publication.

Review form: Reviewer 2

Recommendation

Accept with minor revision (please list in comments)

Scientific importance: Is the manuscript an original and important contribution to its field?

Good

General interest: Is the paper of sufficient general interest?

Acceptable

Quality of the paper: Is the overall quality of the paper suitable?

Good

Is the length of the paper justified?

Yes

Should the paper be seen by a specialist statistical reviewer?

No

Do you have any concerns about statistical analyses in this paper? If so, please specify them explicitly in your report.

No

It is a condition of publication that authors make their supporting data, code and materials available - either as supplementary material or hosted in an external repository. Please rate, if applicable, the supporting data on the following criteria.

Is it accessible?

Yes

Is it clear?

Yes

Is it adequate?

Yes

Do you have any ethical concerns with this paper?

No

Comments to the Author

One of the biggest palaeontological mysteries is the biological affinities of the chitinozoans. This paper adds to the debate by providing evidence that chitinozoans are individual microorganisms rather than the eggs of metazoans (as is most commonly suggested). By adding to this debate this paper will be of interest to palaeontologists and biologists (and indeed the geological community because of the use of chitinozoans in biostratigraphical correlation).

Fundamental to the evidence provided is that the divergent population measured represents a single species (i.e. the variation is intraspecific rather than interspecific). I am not a chitinozoan taxonomist and I suggest that one is consulted as a referee to confirm this critical point.

Furthermore, could the size variation be preservational/taphonomic? This could be through the effects of variation in thermal maturity (that is known to cause size differences in palynomorphs) and/or differences in preparation method (i.e. oxidation) and/or sediment sorting. I think this needs further discussion in the paper.

There is, of course, also the possibility that the extinct chitinozoans represent the egg cases of an extinct group that did display great variation in egg size!

Some specific points:-

Line 76: Were the specimens oxidised in any way? It is important to mention this as oxidation is known to alter the size of palynomorphs.

Line 120: The measured population is from only two samples. Is this enough to cover range in variation?

Line 302: Needs further explanation of why *Desmochitina* is problematic.

Decision letter (RSPB-2019-1270.R0)

25-Jun-2019

Dear Dr Liang:

Your manuscript has now been peer reviewed and the reviews have been assessed by an Associate Editor. The reviewers' comments (not including confidential comments to the Editor) and the comments from the Associate Editor are included at the end of this email for your reference. As you will see, the reviewers and the Editors have raised some concerns with your manuscript and we would like to invite you to revise your manuscript to address them. The reviews are constructive and we hope you agree that the MS will be stronger once they are taken into account.

Research ethics:

Use of animals and field studies:

It is a condition of publication that you make available the data and research materials supporting the results in the article. Datasets should be deposited in an appropriate publicly

available repository and details of the associated accession number, link or DOI to the datasets must be included in the Data Accessibility section of the article (<https://royalsociety.org/journals/ethics-policies/data-sharing-mining/>). Reference(s) to datasets should also be included in the reference list of the article with DOIs (where available).

If you wish to submit your data to Dryad (<http://datadryad.org/>) and have not already done so you can submit your data via this link [http://datadryad.org/submit?journalID=RSPB&manu=\(Document not available\)](http://datadryad.org/submit?journalID=RSPB&manu=(Document%20not%20available)), which will take you to your unique entry in the Dryad repository.

Please submit a copy of your revised paper within three weeks. If we do not hear from you within this time your manuscript will be rejected. If you are unable to meet this deadline please let us know as soon as possible, as we may be able to grant a short extension.

Best wishes,
Professor John Hutchinson
<mailto:proceedingsb@royalsociety.org>

Associate Editor
Board Member: 1
Comments to Author:
Dear Dr Liang:

Thank you for submitting your manuscript to Proceedings of the Royal Society B. I enjoy to read your manuscript and to learn about chitinozoans. Two experts in the field have reviewed your manuscript, and overall, the comments are positive (see below). However, before I can recommend this for publication, all the concerns should be addressed. For example, including in

the discussion, the possibility that "chitinozoans represent the egg cases of an extinct group that did display great variation in egg size".

Best wishes,

Roberto Feuda

Reviewer(s)' Comments to Author:

Referee: 1

Comments to the Author(s)

I revised this paper for another journal, where it was (apparently) rejected.

It is not surprising that some of the chitinozoan workers want to stick to their "egg hypothesis" and do not see this paper in a good light.

However, the paper presents a very well argued study, and with clear evidence they propose another scenario.

My congratulations for a fantastic paper!

I strongly recommend publication.

Referee: 2

Comments to the Author(s)

One of the biggest palaeontological mysteries is the biological affinities of the chitinozoans. This paper adds to the debate by providing evidence that chitinozoans are individual microorganisms rather than the eggs of metazoans (as is most commonly suggested). By adding to this debate this paper will be of interest to palaeontologists and biologists (and indeed the geological community because of the use of chitinozoans in biostratigraphical correlation).

Fundamental to the evidence provided is that the divergent population measured represents a single species (i.e. the variation is intraspecific rather than interspecific). I am not a chitinozoan taxonomist and I suggest that one is consulted as a referee to confirm this critical point.

Furthermore, could the size variation be preservational/taphonomic? This could be through the effects of variation in thermal maturity (that is known to cause size differences in palynomorphs) and/or differences in preparation method (i.e. oxidation) and/or sediment sorting. I think this needs further discussion in the paper.

There is, of course, also the possibility that the extinct chitinozoans represent the egg cases of an extinct group that did display great variation in egg size!

Some specific points:-

Line 76: Were the specimens oxidised in any way? It is important to mention this as oxidation is known to alter the size of palynomorphs.

Line 120: The measured population is from only two samples. Is this enough to cover range in variation?

Line 302: Needs further explanation of why *Desmochitina* is problematic.

Author's Response to Decision Letter for (RSPB-2019-1270.R0)

See Appendix A.

Decision letter (RSPB-2019-1270.R1)

09-Jul-2019

Dear Dr Liang

I am pleased to inform you that your manuscript entitled "Morphological variation suggests chitinozoans may be fossils of individual microorganisms rather than metazoan eggs" has been accepted for publication in Proceedings B. Congratulations!!

Open Access

Your article has been estimated as being 8 pages long. Our Production Office will be able to confirm the exact length at proof stage.

Paper charges

Sincerely,

Professor John Hutchinson
Editor, Proceedings B
mailto: proceedingsb@royalsociety.org

Appendix A

Dear Editor,

We hereby submit the revised version of our manuscript entitled “**Morphological variation suggests that chitinozoans may be fossils of individual microorganisms rather than metazoan eggs**” by Yan Liang and co-authors. We appreciate all the comments and suggestions by the editor and referees.

All the concerns and comments have been carefully considered and addressed. Response to the comments and detailed changes are listed below.

Should you have any further questions, please do not hesitate to contact me at your earliest convenience.

Sincerely,

Yan Liang

Response to Referees

Comments	Answers
Comments from the Associate Editor	
Thank you for submitting your manuscript to Proceedings of the Royal Society B. I enjoy to read your manuscript and to learn about chitinozoans. Two experts in the field have reviewed your manuscript, and overall, the comments are positive (see below). However, before I can recommend this for publication, all the concerns should be addressed. For example, including in the discussion, the possibility that "chitinozoans represent the egg cases of an extinct group that did display great variation in egg size".	The comment is greatly appreciated. It is true that the present study cannot thoroughly exclude the possibility that "chitinozoans represent the egg cases of an extinct group that did display great variation in egg size" since the extinct groups are hypothetical. However, we think that this is an extremely low probability. Although the possibility is extremely low, some readers may have a similar concern. In order to better address this concern, we add two short paragraphs (Lines 305-323) in the Conclusion section to make the concern more clear and also to provide insights for further exploring the nature of chitinozoans. In the first paragraph, we would like to note that the uniformitarianism is of course a basic concept for palaeontology. Much of our knowledge of geologic materials, features and past event is based on the observation of currently active processes. Our science (and Darwinian natural selection itself) is built on uniformitarianism. Unless there is evidence to the contrary, it would be odd to think that ancient eggs had substantially different variation patterns from modern ones. Currently, we didn't find any evidence to the contrary, however, except the great variation, we did find some distinct morphological features which were inconsistent with the egg hypothesis and they were discussed in the Discussion 4.3. In the second paragraph, we discuss the previously reported abnormal specimens, which potentially is the key material recording the reproductive process of chitinozoans. Of course, to demonstrate the reproductive process, systematic arguments

	and new techniques on new materials are needed. Further studies are in progress now and will not be included here. The aim of the present paper is trying to put forward a more reliable possibility to recognize the nature of chitinozoans, rather than simply clinging on the previous egg hypothesis.
Comments from Referee 1	
I revised this paper for another journal, where it was (apparently) rejected. It is not surprising that some of the chitinozoan workers want to stick to their "egg hypothesis" and do not see this paper in a good light. However, the paper presents a very well argued study, and with clear evidence they propose another scenario. My congratulations for a fantastic paper! I strongly recommend publication.	Thank you for your recognition! We will continue the study to further explore the nature of chitinozoans.
Comments from Referee 2	
One of the biggest palaeontological mysteries is the biological affinities of the chitinozoans. This paper adds to the debate by providing evidence that chitinozoans are individual microorganisms rather than the eggs of metazoans (as is most commonly suggested). By adding to this debate this paper will be of interest to palaeontologists and biologists (and indeed the geological community because of the use of chitinozoans in biostratigraphical correlation). Fundamental to the evidence provided is that the (1) divergent population measured represents a single species (i.e. the variation is intraspecific rather than interspecific). I am not a chitinozoan taxonomist and I suggest that one is consulted as a referee to confirm this critical point. Furthermore, (2) could the size variation be preservational/taphonomic? This could be	(1) Yes, it is of great importance to show that the variation is intraspecific rather than interspecific. As we demonstrated in the text, “All the measured parameters are normally distributed and the specimens on the Principal Component Analysis are well grouped, which support the idea that all the specimens belong to a single population”. The morphological changes are also continuous, thus it is reasonable to believe that the divergent population measured represents a single species. (2) Thank you for your concern. All the specimens are in 3D preservation with a series of morphological changes. The variation is with regularity both in size and morphological characters as demonstrated in the text, which has little chance to be affected by the thermal maturity or preservational issues. The preparation methods is following the standard palynological techniques or digested in

through the effects of variation in thermal maturity (that is known to cause size differences in palynomorphs) and/or differences in preparation method (i.e. oxidation) and/or sediment sorting. I think this needs further discussion in the paper. There is, of course, also (3) the possibility that the extinct chitinozoans represent the egg cases of an extinct group that did display great variation in egg size!	diluted acetic acid, both of which have not been reported to cause size differences. To say the least, the specimens were collected from a small piece of rock, around 50 g or 300 g, even if there were some factors which might cause a size difference, the effect should be the same, i.e., the specific value of the specimens might be altered but the size variation of the assemblage should be the same. Moreover, the size variation is not only observed in one species but also can be widely recognized in the previously reported data as illustrated in the text. Therefore, we think that the size variation presented in the study is reliable and can represent the original status. Minor changes are made in Lines 81, 85 and 86 to make the concerns more clear. (3) Please check the response to the editor above.
Line 76: Were the specimens oxidised in any way? It is important to mention this as oxidation is known to alter the size of palynomorphs.	The samples were not processed for any particular oxidation. As explained above, no oxidation which may alter the size of chitinozoans has been detected in chitinozoans following the standard techniques. A short explanation as “without any further process for oxidation” is added in Line 81 to make it more clear.
Line 120: The measured population is from only two samples. Is this enough to cover range in variation?	We are sorry to cause the confusion. The new species were recorded from sample HWY99-37.8 to HWY99-51, in total of six samples. However, the other samples yielded less abundant specimens which were not enough to test the real variation. Further explanation is added in Lines 78, 119 and 122 to make this issue more clear. Additionally, as we pointed out in the Discussion 4.1, the variation in chitinozoans may have been underestimated. With the increasing of the measured specimens, the variation may increase significantly when

	measured specimens were more than 40. For the case study in H. violana , more than 300 specimens are measured. The variation should be reliable.
Line 302: Needs further explanation of why Desmochitina is problematic.	Sorry to cause the confusion. We did not mean to infer that the Desmochitina is problematic, but unique compared to other chitinozoans which carrying a distinct neck and prosome. The reason that why they were unique has been explained in the third paragraph of Discussion 4.3 . The word “problematic” was changed to “unique” in Line 303.